# Proteomics of *Vespa velutina nigrithorax* Venom Sac Queens and Workers: A Quantitative SWATH-MS Analysis

**DOI:** 10.3390/toxins15040266

**Published:** 2023-04-03

**Authors:** Manuela Alonso-Sampedro, Xesús Feás, Susana Belén Bravo, María Pilar Chantada-Vázquez, Carmen Vidal

**Affiliations:** 1Fundación Instituto de Investigación Sanitaria de Santiago de Compostela (FIDIS), Hospital Clínico, 15706 Santiago de Compostela, Spain; manuela.alonso.sampedro@sergas.es (M.A.-S.); xfeassanchez@docente.ucarlemany.com (X.F.); sbbravo@gmail.com (S.B.B.); mariadelpilarchantadavazquez@gmail.com (M.P.C.-V.); 2Research Methods Group (RESMET), Health Research Institute of Santiago de Compostela (IDIS), Network for Research on Chronicity, Primary Care, and Health Promotion (RICAPPS-ISCIII/RD21/0016/0022), University Hospital of Santiago de Compostela, 15706 Santiago de Compostela, Spain; 3Universitat Carlemany, Av. Verge de Canòlich, 47 AD600 Sant Julià de Lòria, Andorra; 4Academy of Veterinary Sciences of Galicia, 15707 Santiago de Compostela, Spain; 5Proteomic Unit, Health Research Institute of Santiago de Compostela (IDIS), University Hospital of Santiago de Compostela, 15706 Santiago de Compostela, Spain; 6Allergy Department, University Hospital of Santiago de Compostela, 15706 Santiago de Compostela, Spain; 7Department of Psychiatry, Radiology, Public Health, Nursing and Medicine, Faculty of Medicine, University of Santiago de Compostela (USC), 15782 Santiago de Compostela, Spain

**Keywords:** Asian hornet, *Vespa velutina nigrithorax*, *Vespa* spp., proteomics, SWATH-MS, hymenopteran, venom, allergy

## Abstract

Health risks caused by stings from *Vespa velutina nigrithorax* (VV), also known as the yellow-legged Asian hornet, have become a public concern, but little is known about its venom composition. This study presents the proteome profile of the VV’s venom sac (VS) based on Sequential Window Acquisition of all Theoretical Mass Spectra (SWATH-MS). The study also performed proteomic quantitative analysis and examined the biological pathways and molecular functions of the proteins in the VS of VV gynes (i.e., future queens [SQ]) and workers [SW]. The total protein content per VS was significantly higher in the SW than in the SQ (274 ± 54 µg/sac vs. 175 ± 22 µg/sac; *p* = 0.02). We quantified a total of 228 proteins in the VS, belonging to 7 different classes: *Insecta* (*n* = 191); *Amphibia* and *Reptilia* (*n* = 20); *Bacilli*, *γ-Proteobacteria* and *Pisoniviricetes* (*n* = 12); and *Arachnida* (*n* = 5). Among the 228 identified proteins, 66 showed significant differential expression between SQ and SW. The potential allergens hyaluronidase A, venom antigen 5 and phospholipase A1 were significantly downregulated in the SQ venom.

## 1. Introduction

Stinging (aculeate) wasps account for approximately 33,000 species across 22 families [1]. Although universally despised, there is evidence that wasps provide similar regulatory, provisioning, supporting and cultural ecosystem services as other insects such as bees [1]. Approximately 1000 species of aculeate wasps are social, belonging to the vespid subfamilies: Polistinae, Stenogastrinae and Vespinae.

*Vespa* Linnaeus, 1758 (Hymenoptera: Vespidae: Vespinae) is one of four genera in the subfamily Vespinae and contains 22 species of hornets [2]. Hornets prey on a wide variety of arthropods and insects, and several hornet species are prolific honeybee hunters. In turn, hornets are preyed on by a variety of natural enemies and have evolved some defense mechanisms, including a coordinated stinging response, potent venom and aggressiveness [3].

*Vespa sp.* venom contains a large and broad array of compounds. The types and concentrations of proteins in hornet venom could differ from one species to another. Hornet venom mainly contains three groups of components: (1) low-molecular-mass peptides (mastoparans, chemotactic peptides and kinins), (2) high-molecular-mass proteins (hyaluronidases, phospholipases, antigen 5, serine proteases and dipeptidyl peptidase IV) and (3) other minor components (acetylcholine, histamine, serotonin, adrenaline, norepinephrine, dopamine and alarm pheromones) [4,5]. In recent years, wasp venom has been the subject of intensive studies on its biological and pharmacological (e.g., antimicrobial, anticoagulant, genotoxic and anti-inflammatory) properties [6]. Some wasp venom compounds have been reported as having beneficial effects in preventing illnesses such as rhinitis, rheumatoid arthritis, ischemia stroke, Parkinson’s disease, Alzheimer’s disease and epilepsy (reviewed in [7]).

In the last two decades, non-native hornet species such as *Vespa bicolor*, *Vespa orientalis*, *Vespa tropica*, *Vespa mandarinia* and *Vespa velutina* (VV), have been sighted outside of their natural range [8,9,10,11,12,13,14,15,16,17,18,19]. Regarding VV, its invasion has become a public health concern in several countries including Spain [8,9,10], France [11,12,13] Portugal [14,15], Italy [16], China [17] and South Korea [18,19] because of allergic and toxic reactions caused by its stings. Recently, the entomological and allergological characteristics of VV have been extensively reviewed by Vidal [20]. Due to its ecology, abundance and wider distribution [21,22], VV represents a major human and animal health risk compared with other native species of Hymenoptera [23,24,25].

Despite the relative lack of research on VV venom toxins, several protein components [26,27,28,29] and other volatile compounds [30,31,32] have been isolated from the venom and characterized. From an allergological point of view, Vesp v 5 and glycosylated Vesp v 1 are relevant allergens in VV anaphylaxis [8,33,34,35]. Preliminary clinical and immunological results in VV allergy had also shown a pattern of sensitization [34], consistent measurement of sIgE and the basophil activation test [35], and the efficacy of *Vespula* venom immunotherapy for treating patients with VV allergy [36]. However, the choice of immunotherapy could be complicated by double vespid sensitization [37].

Recent reviews have shown that hymenopteran venom is a rich cocktail of peptic and proteins with qualitative and quantitative intraspecies and interspecies variation [38,39]. The VV transcriptome in the venom gland has been analyzed and includes 293 putative toxin-encoding sequences, with the two largest families being the hemostasis-impairing toxins and the neurotoxins [40,41]. However, transcriptomics might not fully reflect the final amount of protein compounds in the venom sac. In addition, slight differences have been observed between winter and summer VV venoms using Liquid Chromatography–Mass Spectrometry (LC–MS) combined with multivariate analysis [42].

The aim of this study was to identify proteins in the venom sacs of VV. To the best of our knowledge, this is the first study to use sequential window acquisition of all theoretical mass spectra (SWATH-MS) as a protein quantification tool for the VV venom sac. SWATH-MS data acquisition combines dependent and independent data approaches to simultaneously identify and quantify proteins. We describe the distribution for the various identified protein classes (*n* = 228), as well as their variability in the VV venom sac castes: VV gynes (i.e., future queens [SQ]) and workers [SW]).

## 2. Results

### 2.1. Morphology of the Venom Sac of Vespa velutina Queens and Workers

An adult population of 163 VV (13% males, 63% workers and 24% gynes) was found. As can be seen in Figure 1A, both the morphology and the size of the SW and SQ differ with the SW being longer and more transparent than the SQ, which are more rounded and opaque. These differences result in significantly larger total protein content obtained per venom sac (Figure 1B) in SW than in SQ (274 ± 54 μg/sac vs. 175 ± 22 μg/sac; *p* = 0.02).

### 2.2. SDS-PAGE Analysis of the Venom Sac Potential Allergenic Proteins of Vespa velutina Queens and Workers

The electrophoretic separation of all proteins from the SQ and SW is shown in Figure 1C. The protein-banding pattern of the SQ and SW differs strongly. Clear and distinct bands corresponding to the three main allergens of VV (hyaluronidase, two isoforms of phospholipase A1 and venom antigen 5) can be identified in the SW (SW 1–4), being less prominent in the venom samples of SQ (SQ 1–4).

### 2.3. Proteomic Quantitative Analysis, Biological Pathways and Molecular Functions of Dysregulated Proteins in the Venom Sac of *Vespa velutina* Queens and Workers

To further study the differences in the electrophoretic protein profiles, the collected pooled venom sac was analyzed by LC-MS/MS mass spectrometry, and a quantitative analysis was performed to identify the proteins with differential expression in the venom sac between queens and workers. A total of 228 proteins were quantified in the SQ and SW using the SWATH-MS quantification method. The quantified proteins (Table 1) belong to 7 different classes and 23 different species comprising: (i) mites (*n* = 1), (ii) viruses (*n* = 1), (iii) snakes (*n* = 1), (iv) frogs (*n* = 2), (v) bacteria (*n* = 8) and (vi) insects (*n* = 10). The individual values of SWATH-normalized areas per protein and sample and the fold change (FC) and *p*-values of the *t*-test are available in Appendix A.

Among the 228 identified proteins, 66 showed significant differential expression between SQ and SW samples (FC ≥ 1.5 and *p* < 0.05). Among them, 6 were downregulated and 60 were upregulated in the SQ. These findings support the differences in the electrophoretic profiles of the SQ and SW shown in Figure 1C. A volcano plot (Figure 2) was employed to represent the global quantification of the venom sac proteins in the queens and workers and indicate the dysregulated proteins between the groups.

Among the 60 proteins upregulated in the SQ, the 4 most overexpressed are LIM domain-binding protein (FC = 61.9; *p* = 0.004), 4.5 LIM domains protein 2 (FC = 28.6; *p* = 0.009), phosphoenolpyruvate carboxykinase (FC = 16.5; *p* = 0.002) and transgelin (FC = 16.5; *p* = 0.000).

Regarding the six downregulated proteins in the SQ, we observed decreased expression of beta-galactosidase (FC = 83.3; *p* = 0.008), venom antigen 5 (FC = 19.6; *p* = 0.004), zinc finger protein-like 1 homolog (FC = 3.6; *p* = 0.043), alpha-galactosidase (FC = 3.4; *p* = 0.020), alpha-glucosidase (FC = 2.8; *p* = 0.026), hyaluronidase A (FC = 2.3; *p* = 0.032) and phospholipaseA1 whose expression was 113-fold lower in the SQ than in the SW (*p* = 0.08).

Figure 3 shows the most common gene ontology (GO) terms of the differentially expressed proteins in VV SQ and SW, allowing for a quick comparison of protein functions at the molecular level (Figure 3B) and an assessment of the biological processes (Figure 3A) in which they are involved. The results revealed the involvement of the SW overexpressed proteins in the regulation of carbohydrate and lipid metabolic process, defense response and intracellular cholesterol transport, as well as the participation of the SQ overexpressed proteins in the glycolytic process, phosphorylation, gluconeogenesis, proteolysis, endocytosis and cell differentiation. A detailed list with GO terms is available in Appendix A.

### 2.4. Proteomic Quantitative Analysis from Potential Vespa velutina Allergens

Five of the seven known potential allergens from VV annotated in the toxin database ToxProt (available online: http://www.uniprot.org (accessed on 10 February 2023)) were identified. As shown in Figure 4, hyaluronidase A, venom antigen 5 and phospholipase A1 were significantly downregulated in the SQ (Mann–Whitney test, *p* < 0.05). There were no changes in the expression of hyaluronidase B (FC = 1.12; *p* = 0.570) and phospholipase A1 verotoxin-2b (FC = 2.51; *p* = 0.270).

### 2.5. Proteomic Quantitative Analysis from Dysregulated Proteins of the Class Insecta

The quantified proteins from the class *Insecta* belong to 10 different species (Table 1), seven of them to the Hymenoptera/Apidae lineage. A total of 47, 49 and 82 proteins from *Dufourea novaeangliae, Apis cerana cerana* and *Frieseomelitta varia*, respectively, were quantified in the SQ and SW (Table 2). Overall, 42%, 26% and 29% of proteins from *D. novaeangliae, A. cerana* and *F. varia*, respectively, showed differential expressions between the two groups. For *D. novaeangliae* (*n* = 20); these proteins were upregulated in the SQ. For *A. cerana* (*n* = 13), while 11 proteins were also upregulated in the SQ, two proteins were downregulated. Similarly for *F. varia* (*n* = 24), 22 proteins were upregulated and 2 were downregulated in the SQ.

Among the identified proteins from other species such as *Apis mellifera*
*ligustica*, *Apis mellifera*, *Bombus festivus*, *Xylocopa caerulea*, *Megachile rotundata* and *Bombus humilis* (Figure 5), the proteins phosphoenolpyruvate carboxykinase (FC = 16.56; *p* = 0.002) and 14-3-3 zeta (FC 2.91; *p* = 0.028) were significantly upregulated in the SQ (Mann–Whitney test, *p* < 0.05). Arginine kinase showed upregulation in the SQ (FC 1.84; *p* = 0.057) without reaching statistical significance.

### 2.6. Proteomic Quantitative Analysis from Dysregulated Proteins of the Classes Amphibia, Arachnida and Reptilia

A total of 17 proteins from the class Amphibia in the SQ and SW were quantified, of which 16 belonged to the mimic poison frog *Ranitomeya imitator* and one to the poison dart frog *Dendrobates auratus* (Figure 6D). The hypothetical mimic poison frog protein (Figure 6B) is upregulated (FC 8.01; *p* = 0.001) in the SQ. None of the other 15 proteins from *R. imitator* showed statistically significant differences in expression (Mann–Whitney test, *p* < 0.05). As shown in Figure 6A, five proteins from the mite *Tropilaelaps mercedesae* (class Arachnida) and three proteins from the viper *Bothriechis nigroviridis* (class Reptilia) Figure 6C were quantified in the SQ and SW. For *T. mercedesae*, none of the proteins showed statistically significant differences in expression (Mann–Whitney test, *p* < 0.05). Thioredoxin domain-containing protein (FC = 5.3; *p* = 0.002) and triosephosphate isomerase (FC 2.6; *p* = 0.02), both from the viper *Bothriechis nigroviridis*, were significantly upregulated in the VV queens (Mann–Whitney test, *p* < 0.05).

### 2.7. Proteomic Quantitative Analysis of Dysregulated Proteins from the Classes Bacilla, γ-Proteobacteria and Pisoniviricetes

No differences were observed in the protein expression levels between the SQ and SW from the classes Bacilla (D and E), *γ*-Proteobacteria (A–C and F) and Pisoniviricetes (G) (Figure 7).

## 3. Discussion

A total of 228 proteins were quantified in the SQ and SW, belonging to seven different classes: Insecta (*n* = 191); Amphibia and Reptilia (*n* = 20); Bacilli, γ-Proteobacteria and Pisoniviricetes (*n* = 12); and Arachnida (*n* = 5). The VV is a venomous insect that can inject its biological toxins and proteins into another creature by stinging or spraying, resulting in injury [11,43].

Proteins from γ-proteobacteria, Klebsiella spp., Pantoea agglomerans, Candidatus Schmidhempelia bombi str. Bimp, Gilliamella apis and Pseudomonas spp., Bacilli (Paenibacillus larvae subsp. larvae, Lactobacillus spp., Lactobacillus bombicola) and Pisoniviricetes (deformed wing virus, DWV) were identified in the VV venom sac of both VV queens and workers. The gut bacterial compositions of the genus Vespa from various species and regions have been described in a very similar manner at the phylum and class level [44]. Pantoea agglomerans [45], Gilliamella apis [45,46] and Lactobacillus spp. [44,45], which are the main species in the gut of honeybees, are also present in VV. Klebsiella spp. [44,46] and Pseudomonas spp. [47] have also previously been reported in VV specimens. Despite the generally held view that venom is sterile, microorganisms can viably colonize venoms of vertebrates and invertebrates [48]. The venom sac is connected to the tip of the sting via a persistently open duct exposed to the environment, being comparable to clinical catheterization assemblies: a transcutaneous needle (the sting) that rests on an unsterile surface and is linked to an ongoing duct that leads to a vessel (venom sac) that contains liquid [48]. The analyzed samples could contain not only proteins from the venom itself but also from proteins released from the sac during the procedure. The microorganisms delivered through the sting differ between honeybees, wasps and hornets, with a potentially greater microbial risk for wasp stings in humans [49]. There is an emerging field for integrating microbiology as part of venomics (i.e., venom–microbiomics) for exploring venom as a microenvironment [50]. However, more studies are needed on the origin of these microorganisms, some of which are typically present in the hornet gut flora and which might be present in the extracts analyzed due to cross-contamination during the process of obtaining the sacs.

Regarding the proteins of *Paenibacillus larvae subsp. larvae* and genome polyprotein from DWV (Iflaviridae), it should be stated that *P. larvae* is the etiological agent of American foulbrood, the most dangerous brood disease for honeybees. The species *P. larvae* comprises four different genotypes named ERIC I to ERIC IV. The UniProt V9W8E0 quantified in the present study corresponds to ERIC II genotype, the most important genotype with the highest virulence and one that is frequently isolated from American foulbrood-diseased honeybee colonies [51,52]. To the best of our knowledge, this is the first study to report the presence of *P. larvae* in VV sacs. *Vespa orientalis* has already been shown as a potential reservoir of *P. larvae* [53]. DWV is now the most prevalent pathogen for bee species across the world, in particular *Apis mellifera*. DWV has also been found in 65 arthropod species from eight insect orders and three Arachnida orders [54]. The first report of DWV in invasive VV populations from the Iberian Peninsula since its description was in France (2008 and 2014) [55,56] and in Italy (2017) [57].

Proteins were quantified by SWATH-MS from (i) *Ranitomeya imitator* (*n* = 16), the mimic poison frog, which is naturally distributed in the north-central region of eastern Peru; (ii) *Dendrobates auratus* (*n* = 1), the green and black poison dart frog; and (iii) *Bothriechis nigroviridis (n = 3)*, the black-speckled palm pit viper, native to central America. *R. imitator* mimics not one but three other *Ranitomeya* species of highly toxic poison frogs (*R. fantastica*, *R. variabilis* and *R. ventrimaculata*) [58]. Whether these proteins represent a potential risk for humans after VV stings deserves further analysis.

A total of 191 proteins belonging to class Insecta from a total of ten species were quantified by SWATH-MS in the SQ and SW (Table 1). Previous studies suggest that the entire extant *Hymenoptera* lineage might be descended from a “common venomous ancestor” [59].

Finally, from an allergological point of view, there were several reasons for performing the present study. First of all, allergists face patients with anaphylaxis after being stung by VV which was an unknown specimen involved in such allergic reactions until 2015 [8]. Extensive studies were designed and performed from that moment onwards to find out which allergens were responsible for these reactions and phospholipase A1 (the so-called Vesp v 1 allergen) and antigen 5 (Vesp v 5 allergen) were identified [8,33,34,35,36]. The fact that in venoms from both SQ and SW, Vesp v 1 and Vesp v 5 were isolated was in accordance with our previous studies in which specific IgE against both allergens was detected [8,33,34,35,36]. The larger amount of these allergens in SW could be of importance since the probability of being stung by a worker is much higher than that of being stung by a queen. Such differences could represent an evolutionary composition of the venom in workers to better protect the colony. Even though there are no reports on the allergenicity of hyaluronidase A or B in VV, they would seem to behave as allergens since hyaluronidases from other Hymenoptera have proven their capacity to induce specific IgE in patients [33,34,37]. Unexpectedly, while hyaluronidase A was found in a larger amount in SW, no differences were found between hyaluronidase B in SW and SQ. The meaning of this finding is uncertain.

Another relevant issue to consider in the field of allergy is the convenience of performing sting challenge tests to follow up the efficacy of allergen immunotherapy (reviewed in [60]). Since its first application, it has been used to monitor the time course of protection in patients treated with venom immunotherapy. It implies the use of live insects to cause a real sting in an allergic patient to check the reaction provoked by the venom. Due to ethical issues and since the publication of the transcriptomic of VV venom, identifying several toxins [40,41], a serious concern about inflicting severe reactions during the procedure arose and that was one of the main reasons why we decided to analyze the protein content of VV venom sacs. Thus, previous transcriptomic results suggested a high number of putative toxins in sacs venom of VV. However, transcriptomic analysis involves the study of all the messenger RNA (mRNA) molecules present in a living organism but it does not imply the actual presence of the proteins themselves. It is true that we are 100% sure that contamination with some proteins from the sac itself exists but the methods used in the present study were similar to those employed in the reports of the transcriptomic analysis [40,41]. The fact that not a great amount of toxins has been detected in our venom samples could be of importance to deciding the convenience of the sting challenge test with VV.

## 4. Conclusions

This is the first characterization and quantification of the proteome profile of the venom sacs of *Vespa velutina nigrithorax.* Differences in the electrophoretic profiles of the venom of SQ and SW, both qualitative (presence/absence) and quantitative (intensity of specific bands), have been demonstrated which could have medical relevance. Among the identified proteins (*n* = 228), 29% showed significant differential expression between SQ and SW. The allergen proteins hyaluronidase A, venom allergen 5 and phospholipase A1 were significantly downregulated in the SQ, which should be considered when characterizing the living organism’s response to a VV sting. Relevant differences were found with respect to the putative toxins suggested by the previous transcriptomic analysis, implying a safer scenario when dealing with allergic patients.

## 5. Materials and Methods

### 5.1. Source of Insects

A nest of the yellow-legged Asian hornet, VV was obtained on November 2021 in Galicia (NW Spain). Insects’ castes were identified using their external morphological characteristics [61,62]. A detailed description of the sampling area, nest architecture and identification of VV caste individuals are found in Appendix A.

### 5.2. Venom Collection

After sex and female castes differentiation, the venom sac was extracted from frozen insects by pulling the stinger apparatus from the hornet abdomen by forceps. The venom sac was dissected from the stinger. The venom of 3 sacs (worker or queen) was pooled (4 pooles were obtained) and homogenized in PBS (phosphate-buffered saline). Residual tissues were removed by centrifugation at 3000 rpm × 3 min. The supernatant was moved to a new tube and stored at −20 °C. The venom proteins were quantified by the Bradford method (Bio-Rad, Hercules, CA, USA). The venom proteins were analyzed by 12% SDS-PAGE (sodium dodecyl sulfate polyacrylamide gel electrophoresis) under reducing conditions and Sypro Ruby-stained (Bio-Rad, Hercules, CA, USA). The protein bands were visualized using the ChemiDoc MP imaging system (Bio-Rad, Hercules, CA, USA).

### 5.3. Sample Preparation for Mass Spectrometric Analysis

In relation to tryptic digestion for mass spectrometry, 24 μg protein was concentrated in an SDS-PAGE single band [63,64] and submitted to manual digestion as described elsewhere [65]. Finally, the peptides were dissolved in 0.1% formic acid for further analysis.

### 5.4. Sequential Window Acquisition of All Theoretical Mass Spectra (SWATH-MS) Quantification

For quantitative proteomic analysis, a hybrid quadrupole-TOF mass spectrometer 6600+ (SCIEX, Framingham, MA, USA) coupled to a micro-liquid chromatography (LC) system Ekspert nLC425 (Eksigen, Dublin, CA, USA) was used. Data were collected using aProteinPilot v.5.0.1, PeakView v.2.2, MarkerView and SWATH Acquisition MicroApp v.2.0 software package (SCIEX, Framingham, MA, USA). A customizer database including Vespa + Vespa velutina + Apis mellifera + venom + toxins Uniprot databases (Available online: https://www.uniprot.org/ (accessed on 2 February 2023)) was employed. The obtained peptide mixtures from sample pools were chromatographed for a total time of 40 min operating with data-dependent acquisition in positive ion mode to build the MS/MS spectral libraries, as previously described [66,67,68,69]. The false discovery rate was set to 1% with a confidence score above 99% [66]. This spectral library was used to create the spectral window acquisition used in the SWATH-MS method. Then, 4 μL from each sample were individually analysed, since SWATH-MS technology does not need sample labeling. The SWATH-MS method is based on repeating a cycle that consists of the acquisition of 100-flight mass spectrometry (TOF MS/M) windows. The quantitative analyses of SWATH-MS are supported by extracted ion chromatograms at both the MS1 and MS2 levels. Proteins with more than 10 peptides and seven transitions were selected for quantification. Any shared or modified peptides were excluded. These advantages of SWATH-MS result from retrospectively targeting fragmentation maps to monitor peptides of interest, as well as extendable spectra and virtual libraries. In combination, results identified by SWATH-MS have greater reproducibility, consistency, and sensitivity. These properties contribute to a superior ability to carry out proteomic quantification in a single profiling experiment, especially with higher sensitivity for identifying low-abundance proteins. A Student’s *t*-test analysis was performed for comparison among the samples based on the averaged area sums of all the transitions derived for each protein. The *t*-test was used to indicate how well each variable distinguishes the two groups (SQ and SW), reported as a *p* value. In the set of differentially expressed proteins (*p* < 0.05), a 1.5-fold increase or 0.66-fold decrease was selected as the cut-off point.

### 5.5. Protein Functional Analysis

The differentially regulated proteins were subjected to functional analysis for biological information related to molecular functions, biological processes, cellular components and protein families. We searched them against the protein databases UniProtKB and toxin database ToxProt (Available online: http://www.uniprot.org (accessed on 10 February 2023)).

### 5.6. Statistical Analysis

Graphic expressions of the comparisons between SWATH normalized area of expressed proteins in the sac venom of queens and workers were made using box plots with median, whiskers min to max, and black dots for the individual values per sample. A Volcano plot was generated by plotting the log2 fold change (FC) for the identified proteins against their corresponding adjusted log10 *p* value. FC indicates up- or downregulated proteins if FC > 1.5. A *p* < 0.05 value was considered statistically significant in all tests. Graphics were performed using the GraphPad Prism software (GraphPad Software, San Diego, CA, USA).

## Figures and Tables

**Figure 1 toxins-15-00266-f001:**
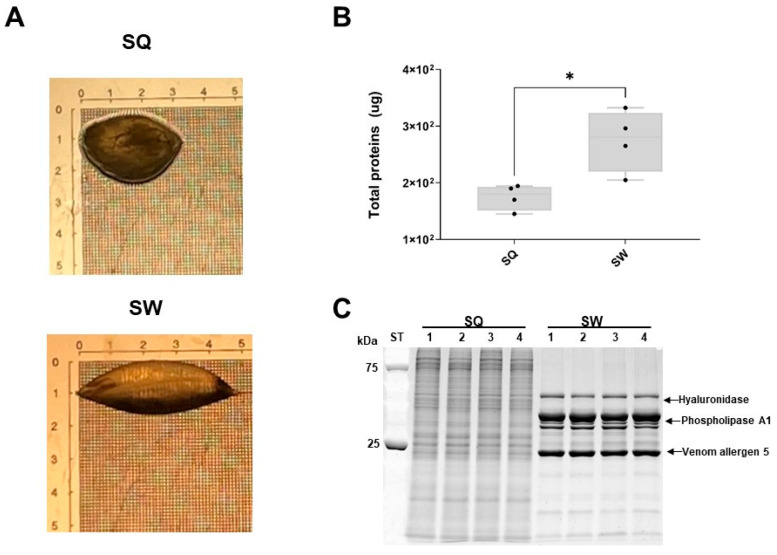
(**A**) Images of the VV venom sac of SQ and SW (scale in mm); (**B**) Box plot with median, whiskers min. to max., and black dots for the individual values per venom sac (* *p* = 0.02); (**C**) Electrophoretic separation of total proteins from SQ and SW. Arrows indicate the bands corresponding to hyaluronidase, phospholipase A1 and venom antigen 5, according to their molecular mass and later confirmed by SWATH-MS. kDa, kilodalton; ST, precision plus protein standard. Bands 1–4 correspond to pooled samples of SQ and SW, respectively. Each pool contains 3 sacs so a total of 12 SQ and 12 SW were analyzed. Photo Author: Manuela Alonso-Sampedro.

**Figure 2 toxins-15-00266-f002:**
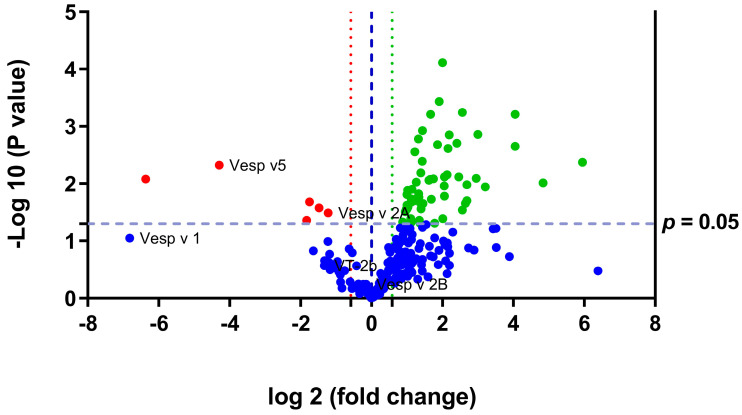
Volcano plot of the venom sac quantitative proteomics data of the VV queens and workers. Volcano plot shows the significantly differentially abundant proteins in the venom sacs by quantitative proteomics analysis. Proteins are ranked in a volcano plot according to their statistical *p*-value (*y*-axis) as—log_10_ and their relative abundance ratio (log_2_ FC) between SQ and SW (*x*-axis). Off-centered spots are those that vary the most between the two groups. The cut-offs for significant changes are FC ≥ 1.5 and *p* < 0.05 (*t*-test). Green spots show the upregulated proteins in SQ, red spots show the downregulated proteins in SQ, and blue spots show the non-dysregulated proteins between the two groups. Vesp v 5, venom antigen 5; Vesp v 1, phospholipase A1; Vesp v 2A, hyaluronidase A; Vesp v 2B, hyaluronidase B; VT 2b, phospholipase A1 verotoxin-2b; SQ, VV queen venom sac; SW, VV worker venom sac. Sample size: SQ = 4 pooled (3 sacs each); SW = 4 pooled (3 sacs each).

**Figure 3 toxins-15-00266-f003:**
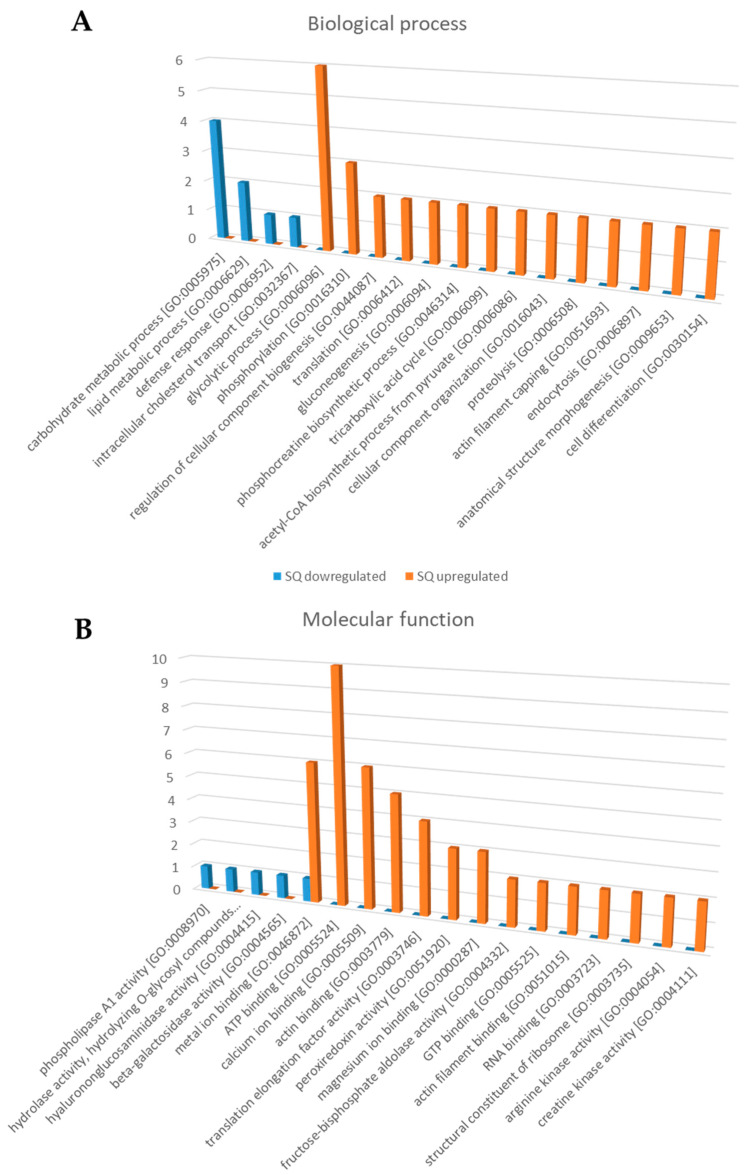
(**A**) Biological processes and (**B**) molecular functions mainly related to the differentially expressed proteins in the venom sac of VV queens (SQ) and workers (SW). The histograms represent the main categories for each gene ontology (GO) term in which differentially expressed proteins were involved (*p* < 0.05). The *y*-axis shows the number of individual proteins in each GO term, and the *x*-axis shows the GO term.

**Figure 4 toxins-15-00266-f004:**
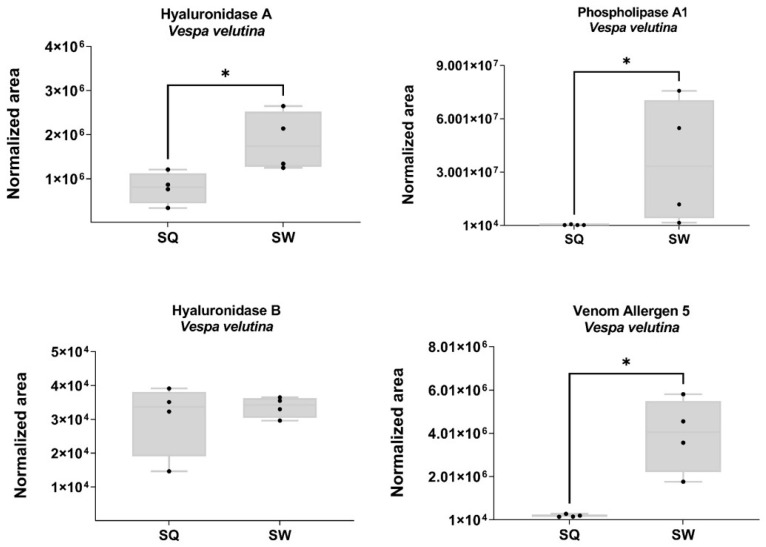
Representation of the SWATH-MS normalized area in potential VV queen and worker allergens. The normalized area was obtained from the SWATH-MS method for each individual sample. Box plot with median, whiskers min to max, and black dots for the individual values per sample. Statistical differences by Mann–Whitney test, * *p* < 0.05. SQ, VV queen venom sac; SW, VV worker venom sac. Sample size: SQ = 4 pooled (3 sacs each); SW = 4 pooled (3 sacs each).

**Figure 5 toxins-15-00266-f005:**
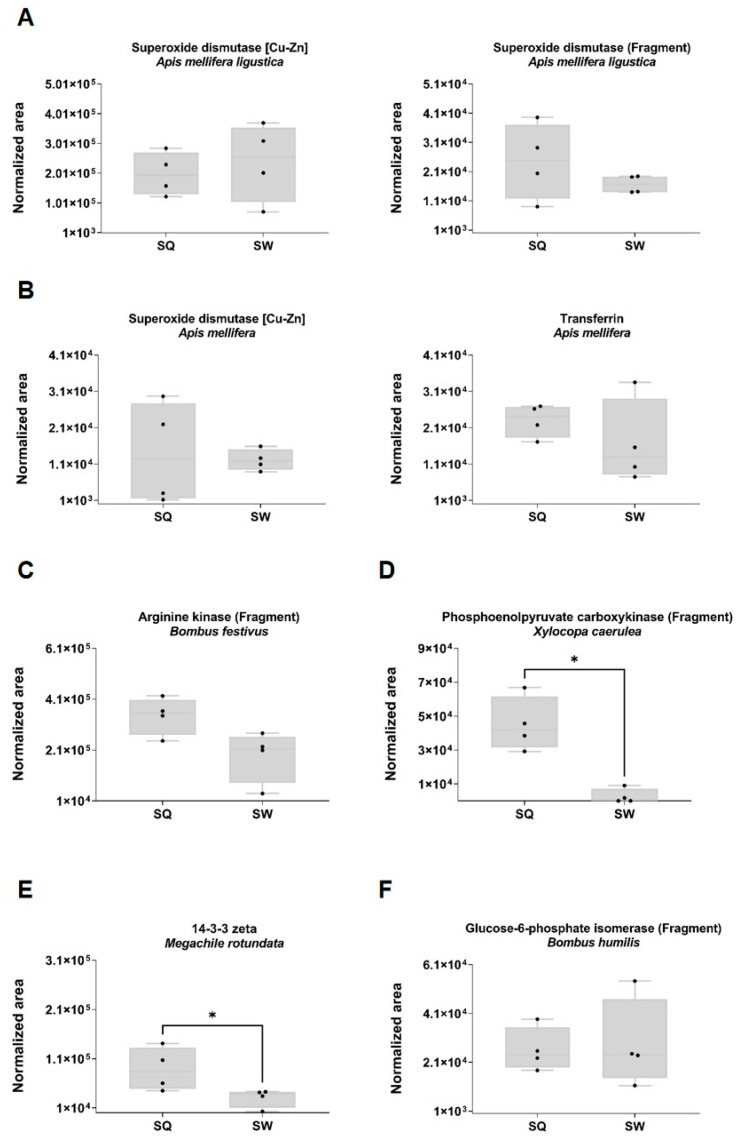
Representation of SWATH-MS normalized area in the SQ and SW quantified proteins from (**A**) *Apis mellifera ligustica*, (**B**) *Apis mellifera*, (**C***) Bombus festivus*, (**D**) *Xylocopa caerulea*, (**E**) *Megachile rotundata* and (**F**) *Bombus humilis*. The normalized area was obtained with the SWATH-MS method for each individual sample. Box plot with median, whiskers min. to max., black dots for the individual values per sample. Statistical differences by Mann–Whitney test, * *p* < 0.05. SQ, VV SW VV SW. Sample size: SQ = 4 pooled (3 sacs each); SW = 4 pooled (3 sacs each).

**Figure 6 toxins-15-00266-f006:**
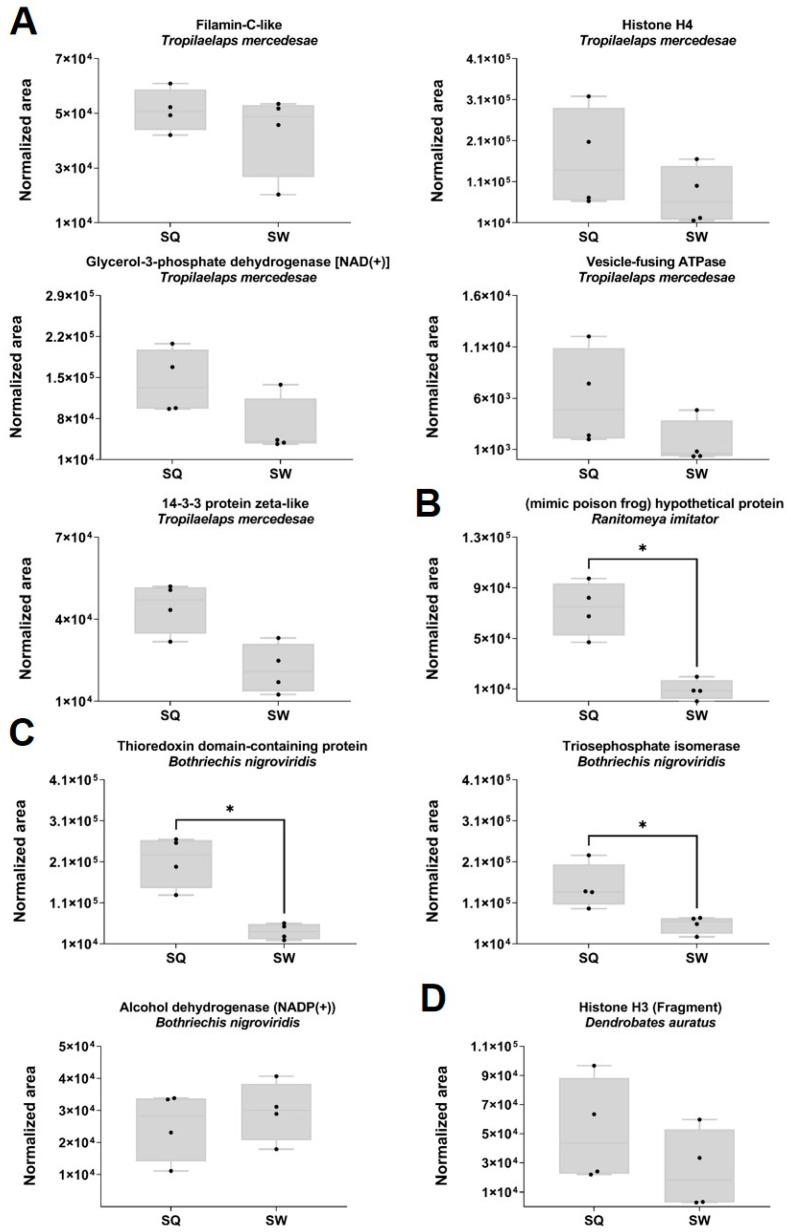
Representation of the SWATH-MS normalized area in the VV SQ and SW quantified proteins from (**A**) *Tropilaelaps mercedesae*; (**B**) *Ranitomeya imitator*; (**C***) Bothriechis nigroviridis*; and (**D**) *Dendrobates auratus*. The normalized area was obtained from the SWATH-MS method for each individual sample. Box plot with median, whiskers min. to max., and black dots for the individual values per sample. Statistical differences by Mann–Whitney test, * *p* < 0.05. Sample size: SQ = 4 pooled (3 sacs each); SW = 4 pooled (3 sacs each).

**Figure 7 toxins-15-00266-f007:**
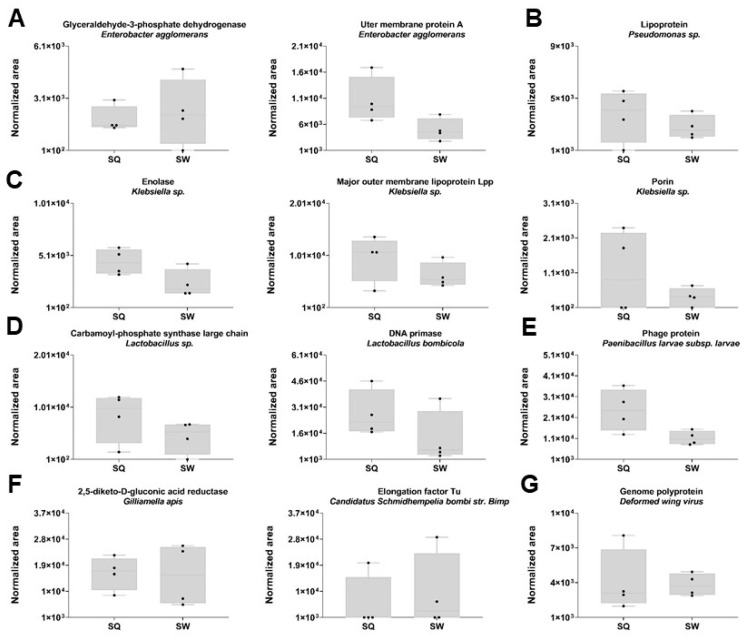
Representation of the SWATH-MS normalized area in *Vespa velutina* (VV) (Hym.: Vespidae) SQ andSW quantified proteins from (**A**) *Enterobacter agglomerans*, (**B**) *Pseudomonas sp. RIT-PI-a*, (**C**) *Klebsiella spp.*, (**D**) *Lactobacillus spp.*, (**E**) *Paenibacillus larvae subsp. Larvae*, (**D**) *Gilliamella apis*, (**F**) *Candidatus Schmidhempelia bombi str. Bimp* and (**G**) *Deformed wing virus*. The normalized area was obtained from the SWATH-MS method for each individual sample. Box plot with median, whiskers min to max, and black dots for the individual values per sample. Sample size: SQ = 4 pooled (3 sacs each); SW = 4 pooled (3 sacs each).

**Table 1 toxins-15-00266-t001:** Classification of the 228 proteins (pp) found in the SQ and the SW by SWATH-MS, according to the species and group to which they belong.

Class	Order/Family	Scientific Name/Common Name	pp	Group
*Amphibia*	*Anura/Dendrobatidae*	*Ranitomeya imitator* (Schulte, 1986)/Mimic poison frog.	16	9NEOB
		*Dendrobates auratus* (Girard, 1955)/Green and black poison dart frog	1	DENAT
*Arachnida*	*Mesostigmata/Laelapidae*	*Tropilaelaps mercedesae* (Delfinado & Baker, 1961)/*Tropilaelaps* mite	5	9ACAR
*Bacilli*	*Bacillales/Paenibacillaceae*	*Paenibacillus larvae subsp. larvae* (White 1906)/American Foulbrood	1	9BACL
	*Lactobacillales/Lactobacillaceae*	*Lactobacillus* sp.	1	9LACO
		*Lactobacillus bombicola* (Praet et al., 2015)	1	9LACO
*γ-proteobacteria*	*Enterobacterales/Enterobacteriaceae*	*Klebsiella* sp.	3	9ENTR
	*Enterobacterales/Erwiniaceae*	*Pantoea agglomerans* (Ewing & Fife, 1972) Gavini et al., 1989	2	ENTAG
	*Orbales/Orbaceae*	*Candidatus Schmidhempelia bombi str. Bimp*	1	9GAMM
		*Gilliamella apis* (Ludvigsen et al., 2018)	1	9GAMM
	*Pseudomonadales/Pseudomonadacea*	*Pseudomonas* sp.	1	9PSED
*Insecta*	*Hymenoptera/Apidae*	*Apis cerana cerana* (Fabricius, 1793)/Asian honeybee	49	APICC
		*Apis mellifera ligustica* (Spinola, 1806)/Italian honeybee	2	APILI
		*Apis mellifera* (Linnaeus, 1758)/European honeybee	2	APIME
		*Bombus festivus* (Smith, 1861)/Bumblebee of Sichuan	1	9HYME
		*Bombus humilis* (Illiger, 1806)/Brown-banded carder bumblebee	1	BOMHU
		*Frieseomelitta varia* (Lepeletier, 1836)/Yellow marmalade bee	82	9HYME
		*Xylocopa caerulea* (Fabricius, 1804)/Blue carpenter bee	1	9HYME
	*Hymenoptera/ Halictidae*	*Dufourea novaeangliae* (Robertson, 1897)/Sweat bee	47	DUFNO
	*Hymenoptera/ Megachilidae*	*Megachile rotundata* (Fabricius, 1787)/Alfalfa leafcutting bee	1	MEGRT
	*Hymenoptera/Vespidae*	*Vespa velutina* (Lepeletier, 1836)/Yellow-legged Asian hornet	5	VESVE
*Pisoniviricetes*	*Picornavirales/Iflaviridae*	*Deformed wing virus*	1	9VIRU
*Reptilia*	*Squamata/Viperidae*	*Bothriechis nigroviridis* (Peters, 1859)/Black-speckled palm pit viper	3	BOTNI

**Table 2 toxins-15-00266-t002:** Proteins from *Dufourea novaeangliae*, *Apis cerana cerana* and *Frieseomelitta varia* with differential expression between VV SQ and SW by SWATH-MS.

Species	Uniprot Code	Protein	*p* Value (*t* Test)	FC (SQ/SW)
*Dufourea novaeangliae*	A0A154P4T7	Four and a half LIM domains protein 2	0.010	28.572
	A0A154NYM3	Transgelin	0.001	16.538
	A0A154PS72	Profilin	0.011	9.233
	A0A154NWX1	Laminin subunit β-1	0.020	6.452
	A0A154PD98	Nidogen-1	0.002	4.470
	A0A154PLJ1	Heat shock 70 kDa protein cognate 5	0.007	4.375
	A0A154PJJ4	Annexin	0.002	3.627
	A0A154PP88	α-1,4 glucan phosphorylase	0.008	3.360
	A0A154PQH0	Glycogenin-1	0.004	2.693
	A0A154P296	Acetyl-CoA hydrolase	0.027	2.644
	A0A154P796	Spectrin α chain	0.006	2.612
	A0A154PAT6	Malate dehydrogenase	0.044	2.552
	A0A154PNQ0	Sortilin-related receptor	0.017	2.535
	A0A154P1L6	ATP synthase subunit β	0.019	2.389
	A0A154P3S2	Muscle M-line assembly protein unc-89	0.003	2.335
	A0A154PJP5	Titin	0.019	2.170
	A0A154PQ65	Multifunctional fusion protein	0.041	2.158
	A0A154P3Q8	Elongation factor 2	0.028	2.003
	A0A154P289	Vacuolar proton pump subunit B	0.016	1.994
	A0A154PPS0	2-phospho-D-glycerate hydro-lyase	0.031	1.930
*Apis cerana cerana*	A0A2A3EBK6	LIM domain-binding protein	0.004	61.974
	A0A2A3EA22	Four and a half LIM domains protein	0.008	7.737
	A0A2A3EL86	Elongation factor 1-γ	0.029	5.917
	A0A2A3EC59	Calpain-A	0.000	3.745
	A0A2A3E2W8	Pyruvate dehydrogenase E1 component subunit α	0.019	3.234
	A0A2A3EGD2	Arginine kinase	0.022	2.724
	A0A2A3EMK6	α -actinin, sarcomeric	0.001	2.719
	A0A2A3EKS3	Fructose-bisphosphate aldolase	0.049	2.583
	A0A2A3E5U7	Talin-2	0.020	2.285
	A0A2A3EPX9	Glutamate dehydrogenase (NAD(p)(+))	0.026	2.042
	A0A2A3E464	ATP synthase subunit α	0.025	2.028
	A0A2A3EAL0	Zn finger protein-like 1 homolog	0.044	0.281
	A0A2A3EJN3	Acid β-galactosidase	0.008	0.012
*Frieseomelitta varia*	A0A833VQ28	40S ribosomal protein S4	0.010	6.475
	A0A833VR32	EF-1-γ-C-terminal domain-containing protein	0.022	6.293
	A0A833S6V3	Neurochondrin homolog	0.001	5.899
	A0A833VVD0	Isocitrate dehydrogenase [NAD] subunit, mitochondrial	0.008	5.509
	A0A833R5C8	Lipocln_cytosolic_FA-bd_dom domain-containing protein	0.001	4.562
	A0A833VMV6	Eukaryotic translation initiation factor 5A (eIF-5A)	0.017	4.163
	A0A833RMB6	Muscle LIM protein Mlp84B	0.008	4.161
	A0A833RMX9	Filamin-A	0.011	4.147
	A0A833W6L4	Small nuclear ribonucleoprotein-associated protein	0.041	4.017
	A0A5P1MU32	Glyceraldehyde-3-phosphate dehydrogenase	0.000	4.000
	A0A833WD27	Pyruvate dehydrogenase E1 component subunit β	0.049	3.428
	A0A833S5I5	Phosphoglycerate kinase	0.001	3.175
	A0A833R5H5	DJ-1_PfpI domain-containing protein	0.008	3.170
	A0A833RCZ1	Thioredoxin domain-containing protein	0.009	3.081
	A0A833S1K4	Malic enzyme	0.021	2.587
	A0A833VV22	WD_REPEATS_REGION domain-containing protein	0.015	2.522
	A0A833W6G6	Pyruvate kinase	0.002	2.501
	A0A833S8P7	ADF-H domain-containing protein	0.009	2.389
	A0A833RUN1	Heat shock 70 kDa protein cognate 4	0.013	2.205
	A0A833W2V9	Fructose-bisphosphate aldolase	0.021	2.173
	A0A833WDK7	Calreticulin	0.024	2.003
	A0A833WFT3	Thioredoxin domain-containing protein	0.028	1.975
	A0A833RKT2	α-glucosidase	0.027	0.358
	A0A833W1X0	α-galactosidase	0.021	0.298

FC (SQ/SW), fold change of queen/worker; FC > 1 overexpressed protein in SQ; FC < 1 downregulated protein in SQ; *p* value < 0.05: statistically significant.

## Data Availability

The mass spectrometry proteomics data have been deposited to the ProteomeXchange Consortium via the PRIDE partner repository with the dataset identifier PXD040764.

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
