# Peer review of "Proteomics of Vespa velutina nigrithorax Venom Sac Queens and Workers: A Quantitative SWATH-MS Analysis"

_toxins, 2023, doi:10.3390/toxins15040266_

Round 1

Reviewer 1 Report

 I have revised the MS "Venom Proteomics of Vespa velutina nigrithorax (Hym.: Vespidae) Queens and Workers: A Quantitative SWATH-MS Analysis". The study is innovative for the methodology used and for the results, which are of great interest to assess the risk for human and animal health connected to the sting of this species. 

The introduction provides a good overview about the topic and the discussion and conclusions fit the results. However, part of the methodology is not clear, such as the number of samples analysed. 

Supplementary material 1 and supplementary material 2 are provided as excell files, this makes it impossible to have heading and other information available to understand what is reported.

I have a few more comments  which should be addressed to improve the MS:

Line 24-25: please provide reference.

Line 27-28: please rephrase.

Line 42: if different studies have provided data underlining the beneficial effects of venom, please provide more reference; or turn the sentence to singular “a previous study…”

Lines 45-46: you should probably also nominate Vespa velutina and you should provide references

Line 51: instead of “habits” would “behavior” “ecology” be more appropriate?

Line 79-80: You state that SW are longer than SQ, are there any numerical data which you can provide?

Figure 1A: Please indicate a scale. Are the numbers indicating cm or mm?

Figure 1C: What are the number 1-4 indicating? Are they samples? If so, why are there 4 samples for SW and 4 samples for SQ if in the sampling paragraph it is stated that only 3 samples were collected? Please verify and specify

L88-90: Should be moved to the materials and methods section.

L113-114: please use SC and SW as in the rest of the text

L322: how was the sac cleaned? Have a clearly understood that 3 samples of SW+ 3 samples of SQ were analysed (total 6 samples)? It also does not fit with Figure 1C or it is not clear.

Reviewer 2 Report

Reference:  “Venom Proteomics of Vespa velutina nigrithorax (Hym.: Vespi- 2 dae) Queens and Workers: A Quantitative SWATH-MS Analy- 3 sis”. Submitted to TOXINS, March, 2023.

General comments: In the work, the authors report their data obtained through proteomic analysis of the venom gland of the wasp Velutina nigrithorax, which is a public health problem in various parts of the world. Mass spectrometry analyses of venom glands of individuals called future queens [SQ]) and workers [SW] within the swarm were performed. In the analyses, the authors found a total of 228 proteins belonging to 7 different classes: Insecta (n=191); Amphibia and Reptilia (n=20); Bacilli, γ-Proteobacteria and Pisoniviricetes (n=12); and Arachnida (n=5). The data also showed variations between the concentration of proteins and some toxins in the venoms of groups [SQ] and [SW. After careful reading, it is my opinion that the topic falls within the scope of TOXINS, the text is clear, and with relevant scientific content. Nevertheless, here are some suggestions that authors can incorporate into a revised version to make the text more attractive and complete.                 

Specific Comments:

1-     Throughout the text, please change molecular weight to molecular mass, because molecules do not have weight, but mass. What has weight is rice, meat, bread,... The authors themselves used mass spectrometry and not weight spectrometry. Molecular weight is a term used in the literature, but incorrectly. Let's make this text more correct.  

2-     Between lines 37 and 38 the authors wrote …. high molecular weight proteins (hyaluronidase, phospholipase, antigen 5, serine protease and dipeptidyl peptidase IV). I would write hyaluronidases, phospholiases, serine proteases in the plural, because there are several isoforms for these molecules.

 3-     Between lines 64 and 66 the authors wrote …. The VV transcriptome in the venom gland has been analyzed and includes 293 putative toxin-encoding sequences, with the two largest families being the hemostasis-impairing toxins and the neurotoxins [40,41]. What is the authors' opinion on the discrepancies between analyzes of VV transcriptomes and proteomes, as discussed? Could samples used in transcriptome or proteome analyzes be contaminated with cellular components that are not toxins?  

 4-     As a suggestion, I would put a photo of male and female representatives of these wasps in the introduction, to facilitate identification by clinicians who are not specialists in the area.

 5-     In the line 91, legend of figure 1, the authors wrote VV after SQ and SW, but as they are working with Vespa velutina venom and already define VV on line 85, they do not need to repeat it in line 91.

 6-     About the electrophoretic profile shown in figure 1C and attributed to proteins (hyaluronidase, phospholipase A1 (both isoforms), and venom antigen 5). In my opinion, it would be interesting if the authors describe the molecular mass of each toxin, in addition to showing a Western Blotting reaction to confirm that they really are the toxins shown. An SDS-PAGE in the first dimension can have lot of co-located molecules.

7-     Also some discussion about the discrepancies shown could be placed in the text. For example, workers (SW) are more subject to combats for predation and protection of the swarm and hence logically have higher concentrations of toxins involved in the defense of the colony.

 8-     Also, in the text, the authors need to define whether all collections were from adult animals, to minimize criticism regarding the variations noted in gland sizes and venom compositions.

 9-     In the line 106, the authors wrote …. using the SWATH-MS quantification method. It would be interesting for the authors to explain here or in M/M what is new about this method compared to previous ones in the literature, which also use MS?

10-  About Table 1, although it is clear and explanatory, in my opinion the authors could show a classification of proteins sequenced and identified by MS, instead of, for example, showing order/family of identified animals. Showing some more direct data on the types of proteins identified is much more important, including for proposing treatments and direct practical arrangements.

 11-  Another questioning of these types of analyses, due to the high sensitivity of the method, is that materials originating from the cells of venom producing glands, but which do not have toxic potential, are also generally sequenced/identified. I would like the authors' opinions on the toxic potential of some proteins that appear in the text between lines 116 to 131. Such as  phosphoenolpyruvate carboxykinase, beta-galactosidase, zinc finger protein-like 1 homolog, alpha-galactosidase, alpha-glucosidase. It seems to me that they are components of cells!

 12-  A table could show the same data of figure 2, but in a simpler, correct and complete way.

 13-  Figure 3 summarizes my comments on question 11. Where the authors themselves comment on possible cellular metabolic functions of the proteins identified in the analyses, suggesting that they are not toxins, but probably from the cells that are part of the glands (cytosolic, nuclear or Matrix extracellular). A more refined analysis would be interesting, separating proteins involved in cellular processes than those that could actually be toxins. There are software’s able to do this!

 14-  About the text of figure 4 , line 164 …. and phospholipase A1 verotoxin-2b (FC = 2.51; p = 0.270). Put data not shown, as these data do not appear in figure 4.

 15-  Still on the text related to figure 4, make it clear why allergenic toxins from the venoms were chosen among several other toxins with harmful potential  .... Proteomic quantitative analysis from potential Vespa velutina allergens (line 159). Probably because allergenic and anaphylactic reactions are related to clinical complications in accident victims. An explanation about this is welcome! 

 16-  About table 2, where it seems clear to me that the vast majority of the proteins shown are involved in cellular metabolic processes, or are structural proteins of the cytoskeleton, of the extracellular matrix, for example, and do not seem to me to have a role in the toxic events described during accidents. Once again, the authors should have made a refinement, trying to separate proteins with possible toxicological functions from non-toxicological ones, and then look for some relationship with the venoms of animals already involved in accidents, such as bees, other wasps, other insects, spiders...  

17-  About figures 5 and 6, where the authors show comparisons of proteins identified in the glands of Velutina nigrithorax, with molecules present in frogs, spiders, reptiles, other insects. Once again, it would be interesting to show the toxicological significance of the identified proteins. Otherwise, they are not from the venom, but structural, cytosolic, general metabolism molecules, among other examples. They are not venom! And if it is not venom what is the meaning for public health and envenoming by Velutina nigrithorax?

 18-  Based on the text described between lines 247 to 252, could the presence of bacterial and viral proteins in the analyzed venom reflect microbial contamination of animals during manipulations of the soil, environment, among others, and thus not be components of the venoms?

 19-  In the text described between lines 268 to 280, the authors comment on the origin of the proteins of Paenibacillus larvae subsp (a plague that affects bees), described in the venoms of Velutina nigrithorax. Are these wasps resistant to the disease, but could they spread these microbes to native bees and put at risk commercially exploited swarms for honey extraction in the Region where the collections were made and even in the country?

 20-  In the line 293 …. (the so-called Vesp v 1allergen), please format to Vesp v1 allergen.

 21-  In my opinion, I found the discussion of the text too small. The final part between lines 292 to 300 was very good, but I missed an analysis where the authors could discuss, among all the identified and sequenced proteins, those that could really be classified as toxins. There are programs capable of doing this from the amino acid sequence. This could bring news to the toxinological picture described for these wasps.

 22-  I also missed, in the discussion of the text, the authors' opinion on several sequenced proteins, which are very likely to be cellular proteins that were extracted during the process of obtaining samples, and are not part of the venom.

 23-  The concluding chapter between lines 301 to 309 was very summarized within everything that was described throughout the text. The authors could include many other data.

24-  In the line 326 please change SDS PSGE by 12% SDS-PAGE.

25-  In the line 348 … Vespa velutina + Apis mellifera + poison + toxins. In my opinion the best word to use here is venom changing the word poison. Poison is a word more appropriated to chemical compounds.

 26-  Between lines 379 to 383, where the authors pointed the methodology used to do Protein Functional Analysis. In my opinion the programs indicated (UniProtKB and 382 toxin database ToxProt http://www.uniprot.org) could be used to select toxins from cellular molecules as argued along of my points.

27-  I would like an in-depth discussion by the authors on the direct applicability of this work and a more direct action by medical practices on patients injured by this wasp, which, as indicated by the authors, is a public health problem in several countries around the world, and justified this work!

28-  It would be interesting for the authors to discuss, among the various proteins sequenced in this work, those that have toxic activities, showing their presence in venoms and indicating to profissionals involved in the care of injured patients, what are their possible toxic effects. This would undoubtedly contribute to the public health!
